# Views of patients suffering from Failed Back Surgery Syndrome on their health and their ability to adapt to daily life and self-management: A qualitative exploration

Tanja E. Hamm-Faber[1]*, Yvonne Engels[2], Kris C. P. Vissers[2], Dylan J. H. A. Henssen[3]

1 Department of Pain Medicine, Albert Schweitzer Hospital, Zwijndrecht, The Netherlands, 2 Department of Anesthesiology, Pain and Palliative Medicine, Radboud University Medical Center, Nijmegen, The Netherlands, 3 Department of Medical Imaging, Radboud University Medical Center, Nijmegen, The Netherlands

* tefaber01@gmail.com

**Data Availability Statement:** All relevant data are within the manuscript and its Supporting Information files.

## Abstract

### Background

The clinical outcomes of Spinal Cord Stimulation (SCS) therapy in patients with a Failed Back Surgery Syndrome (FBSS) is mostly done by standardized pain and quality of life measurements instruments and hardly account for personal feelings and needs as a basis for a patient-centred approach and shared decision making.

### Objectives

The objective of this study is to explore perspectives on personal health and quality of life (QoL) in FBSS patients concerning their physical-, psychological and spiritual well-being prior to receiving an SCS system.

### Methods

We performed face-to-face, semi-structured, in-depth interviews to obtain descriptive and detailed data on personal health, guided by the Web diagram of Positive Health (Huber et al.) and a topic list. The following main topics were assessed qualitatively: 1) Bodily functioning, 2) Mental function and perception 3) Spiritual dimension, 4) Quality of life, 5) Social and societal participation and 6) Daily functioning.

### Results

Seventeen FBSS patients (eight male, nine female) were included from April–November 2019 at the department of pain medicine in the Albert Schweitzer Hospital in the Netherlands. Median age 49 years; range 28 to 67 years, and patients underwent between one and five lumbar surgical operations. The duration of their chronic pain was between four and 22 years. After analyzing the interviews, three themes emerged: 1) dealing with chronic

**Funding:** This is an unfunded study.

**Competing interests:** No, authors have no competing interests.

pain, 2) the current situation regarding aspects of positive health, and 3) future perspectives on health and quality of life. These themes arose from eleven categories and a hundred ninety codes.

## Conclusion

This qualitative study explored FBSS patients 'views on their health and the ability to adapt to daily life having complex chronic pain, and showed that patients experienced shortcomings in daily life within the six dimensions of the Web diagram of Positive Health before the SCS implant.

## Introduction

Patients with Failed Back Surgery Syndrome (FBSS) have had one or more spinal surgical interventions without resulting pain relief for the low back and/ or leg [1, 2]. Instead, during a long medical trajectory, they often not only have developed long-lasting chronic pain but also an important loss of quality of life presented by continuous fatigue, low sleep quality and inability to function normally during their daily activities [3, 4]. Moreover, FBSS patients' social involvement often decreases as it becomes more challenging to participate in group-activities in social life (can be family and friends) when dealing with physical, psychological, social- environmental and occupational barriers [4, 5]. Additionally, these patients experience constraints concerning their mood and depression as they experience an imbalance in their physical challenges. They become more dependent on their environment, and consequently feel frustrated and disappointed, and develop an increasingly reclusive lifestyle [5].

In general, FBSS patients alternately feel hope and disappointment, which both can be derived from proposed medical interventions and eventual treatments. In case of refractory low back pain and/ or leg pain, one of the last resorts for treatment is Spinal Cord Stimulation (SCS) as this therapeutic option has proven evidence on their quality of life and perspective on a better function in daily life, duties and future [2, 6].

It is often questioned how patients with chronic low back pain and leg pain (CLBLP), due to FBSS, participate in a performance-oriented society. To understand and evaluate the complexity of chronic pain of patients receiving SCS therapy, the biopsychosocial model first developed by George Engel (1977), is universally accepted as a model introducing multidimensional aspects in chronic pain syndromes [7]. However, most medical professionals only have limited insights in the quality of life, personal needs, the meaning of life, and coping strategies of their patients, who are confronted with repeated disappointing experiences or how they can self-manage their life given their chronic pain [8].

The dynamic interaction between the physical, psychological and social dimension is specific for each patient and should be multidimensionality managed in relation to individual quality of life and health status with chronic pain. Currently, the evaluation of the clinical outcomes of SCS therapy in patients with CLBLP due to FBSS is mostly done by standardized pain and quality of life measurements instruments, which hardly account for personal feelings and needs, which contrasts with the patient-centred Web diagram of Positive Health of Machteld Huber [8, 9]. This diagram is guided by individual capabilities and self-management, and gives direction for personally tailored treatment as a basis for shared decision making between a patient and healthcare professional to increase the therapy outcome. The Web diagram includes six dimensions: 1) Bodily functioning, 2) Mental function and perception 3) Spiritual

dimension, 4) Quality of life, 5) Social and societal participation and 6) Daily functioning. These dimensions appeared to be in line with what FBSS patients consider essential in daily life [10, 11].

So far, person-centred goals regarding health and the ability to adapt and self-manage in chronic pain have hardly been explored. Earlier studies investigated patients 'experiences after SCS implantation [11], educational needs in FBSS patient [12], and expectations in the period that SCS was planned [10]. One study explored the importance of goal identification before SCS [13].

Therefore, with this qualitative study, we aim to get insights in the views of patients suffering from Failed Back Surgery Syndrome on their health and their ability to adapt and self-manage in daily activities before they received an SCS by exploring their experiences informed by the elements of the Web diagram of Positive Health as proposed by Huber et al.

## Methods and instruments

### Ethics statements

The Medical Ethical Board of the Albert Schweitzer Hospital in the Netherlands approved the study. (Reference number 2019–014). Participants declared that they understood the information and signed the informed consent prior to being scheduled for an interview.

The interviewer (T.H.) is an employer at the Department of Pain Medicine of the Albert Schweitzer Hospital as a Master of Arts in Advanced Nursing Practice (MA-ANP) and had no personal or professional history with the participants who were included in this study.

The interviewer (T.H.) was trained in qualitative research and communication techniques. Furthermore, she is experienced in the management of patients suffering from chronic pain and neuromodulation therapy. We refined the interview skills of the interviewer by discussing audio recorded interviews with an experience interviewer (J.W.), and receiving feedback by two experts in scientific qualitative research. (Y.E. D.H.).

### Participants

Participants were invited to take part in the study for those who experienced limitation in daily life having chronic pain as a result of a Failed Back Surgery Syndrome and meet the inclusion criteria of the Dutch Neuromodulation Society.

The Dutch Neuromodulation Society developed the guidelines for healthcare professionals in the Netherlands, who are directly involved in the neuromodulation therapy in order to monitor the quality of the treatment. Healthcare professionals should follow these rules to be able to proceed the therapy.

Purposive sampling took place, aiming to include a representative distribution of Dutch men and women between 18–75 years, suffering low back pain and leg pain after surgery on the Spine. Sample size continues until saturation is reached, meaning that in the line by line coding process, no new codes emerge. In our study, we performed two extra interviews to confirm data saturation.

All eligible participants were recruited by pain specialists of the department of pain medicine of the Albert Schweitzer Hospital, between April and November 2019 Participants voluntarily participated in this study and received a study number which guaranteed their privacy.

Inclusion criteria were: 1) fluency in the Dutch language 2) age $\geq$ 18 years, 3) diagnosed with FBSS with no option for further surgical interventions, 4) severe pain $\geq$ 5 on a Visual Analogue Scale for more than six months, and 5) no psychopathology (no mental illness or substance abuse) as determined by a psychological assessment. Exclusion criteria for this study were clinical contraindications for SCS (Table 1).

**Table 1. Characteristics of the participants.**

| Patient | Gender | Age | Number of surgical intervention | Duration of pain | EQ5D score | Medication |
|---------|--------|-----|--------------------------------|------------------|------------|------------|
| P.1 | M | 58 years | 3 | 12 years | 4.0 | Oxycodon 2 x 20 mg. |
| | | | | | | NSAIDs 2 x 50 mg. |
| | | | | | | Acetaminophen 2 x 1 gr. |
| P.2 | V | 46 years | 2 | 10 years | 3.5 | Zaldiar (= Tramadol 37,5 mg/ Acetaminophen 12,5 mg) 3 x DD1 |
| P.3 | V | 46 years | 3 | 4 years | 5.0 | Oxycodon 4 x 5 mg. |
| P.4 | M | 36 years | 1 | 7 years | 3.5 | Gabapentin 6 x 400 mg. |
| | | | | | | Tramadol 1 x 50 mg. |
| | | | | | | Acetaminophen 4 x 500 mg. |
| P.5 | M | 45 years | 1 | 22 years | 4.5 | Pregabalin 4 x 75 mg. |
| | | | | | | Fentanyl 25 Mug- 3 days |
| P.6 | V | 40 years | 3 | 4 years | 1.5 | Pregabalin 2 x 75 mg. |
| P.7 | M | 45 years | 1 | 4 years | 4.0 | Pregabalin 2 x 75 mg. |
| | | | | | | Tapentadol 2 x 100 mg. |
| P.8 | V | 55 years | 1 | 6 years | 4.0 | Tapentadol 2 x 50 mg. |
| | | | | | | Acetaminophen 1 x 1 gr. |
| P.9 | V | 63 years | 3 | 7 years | 5.2 | Pregabalin 2 x 75 mg. |
| P.10 | M | 63 years | 1 | 2 years | 3.0 | Tramadol 3 x 50 mg. |
| | | | | | | Acetaminophen 3 x 1 gr. |
| P.11 | V | 28 years | 1 | 7 years | 4.0 | - |
| P.12 | M | 67 years | 1 | 3 years | 5.0 | - |
| P.13 | V | 43 years | 2 | 2 years | 4.0 | Pregabalin 4 x 75 mg. |
| P.14 | V | 43 years | 1 | 3 years | 4.0 | Amitriptyline 1 x 10 mg. |
| P.15 | V | 47 years | 1 | 7 years | 5.0 | - |
| P.16 | M | 46 years | 5 | 7 years | 3.5 | Duloxetine 1 x 60 mg. |
| | | | | | | Oxycodon 2 x 10 mg. |
| P.17 | M | 63 years | 3 | 15 years | 3.0 | Acetaminophen 3 x 1 gr. |

The location of the interview was at a comfortable environment in the hospital or patients own home. The hospital location was an outpatient clinic room with a window view on a park. The green and white coloured room is decorated with modern art, and had two comfortable chairs for the interviewer and interviewee. On the table between them was coffee/tea and cake and the temperature was adjusted to patients 'wishes.

## Design

Face to face, semi-structured in-depth interviews were held to obtain detailed insights into the quality of life of FBSS patients for which we used a the Web diagram of Positive Health and a topic list.

The theoretical part of Positive Health was introduced by Huber et al. in which she stated that health is not the absence of illness but the ability to adapt to physical, emotional and social challenges, self-management and motivation for personal goals. She developed the Web diagram with the six dimensions of Positive Health preferred by the patients, which are: 1) Bodily functioning, 2) Mental function and perception 3) Spiritual dimension, 4) Quality of life, 5) Social and societal participation and 6) Daily functioning [8, 9].

If patients were interested in taking part in the study, the Web diagram was explained, as a model that helps to visualise the broad perception of health aspects and starts the patients'thinking process on the six dimensions of Positive Health. The patient is asked to give a score

from 0 (no quality) to 10 (best quality) for each of the six dimensions. Patients had received a copy to take home during a previous outpatient clinic visit, and they completed this template on the day of the interview. This tool visualises and quantifies the six dimensions of the Web diagram and initiates communication with the patient.

The topic list (S2 File) is used as a guide to structure the semi structured interview, and concerned exploration or matters that patients consider essential in their health and their ability to manage their lives with chronic pain. Furthermore, the topic list served as a guide for the most important topics the interviewer would like to discuss. Each topic had several sub topics. During the interviews, we refined these sub topics. Additional topics, which emerged during the interviews, were added to the topic list.

## Planning and interview

All interviews took place 2–3 weeks before the planned SCS implantation date and were conducted by one researcher, who was trained in qualitative research and interviewing (T.H).

The interviewer conducted the interviews as near as normal conversation by using open-ended questions. Participants were encouraged to express their feelings and to elaborate on aspects of their daily lives and possible daily challenges they encountered on all dimensions. When answers were unclear to the interviewer, additional questions were asked to ensure a clear understanding of the meaning of the patients 'expressions.

The interviews were audio-recorded and transcribed verbatim, and if necessary field notes were made for elements which cannot be recorded.

If any uncertainties would remain after analysing the interviews, the researchers had the opportunity to re-contact the interviewees.

## Data analysis

For the qualitative content analysis, we used the phases of thematic analysis at which we explore an inductive approach using the Web diagram with six dimensions of Positive Health as themes for the semi-structured in-depth interviews.

Starting after the first interview, transcriptions were coded line-by-line, through which a code list was created (S1 File). The code list derived from the previous interview was used as a starting point for coding the next one. Coding was continued after each interview.

Coding and analysis were carried out independently by two researchers (T.H. and D.H.), who met periodically to compare individual coding and to discuss codes and themes until consensus was reached. By employing this methodology, the researchers independently signalled data saturation when no new topics, themes or codes derived when analysing new interviews. After signalling data saturation, two additional interviews were held to ensure no new topics or codes arose.

After all interviews were conducted, transcribed and coded, the research team (T.H, Y.E, K. V. and D.H.) combined codes into categories and themes (S3 File).

The coding was performed using ATLAS.ti. Scientific Software Development GmbH, Berlin. Document version: 673.20190310 (http://atlasti.com).

The transcripts were not returned to the participants for their comments.

## Quality

The study was conducted in accordance with the principles of the Declaration of Helsinski and approved by the Medical Ethical Board. All participants voluntary signed the informed consent and receive a code. This code is not based on patients 'characters to guaranteed their privacy.

In this qualitative study, we used audio-taped data. The transcript were analysed in Atlas.ti, under licensed of the Radboudumc in the Netherlands. All data were anonymously stored at an Investigators Site File at the Radboudumc and only accessible by qualified control persons and research management team. The data will be saved for15 years.

## Results

Each interview lasted between 45–60 minutes. They all took place in a comfortable environment in the hospital, where only the participant and the interviewer were present. Saturation was reached after having interviewed fifteen participants; after which two additional interviews took place to confirm saturation.

Of the seventeen participants, eight were male. The median age was 49 years, ranging from 28 to 67 years. Participants had had between one and five spinal surgeries and the duration of chronic pain was between four and 22 years. None of the recruited participants refused their participation in this study.

After having analysed the interviews, three themes emerged: 1) Dealing with chronic pain, 2) The current situation regarding aspects of Positive Health and 3) Future perspectives on health and quality of life. These themes arose from eleven categories and 190 codes (Table 2).

The majority of participants scored a relative low number on the six axes of the of Web diagram of Positive Health which they filled in at the start of the interview (Fig 1).

## Theme 1: Dealing with chronic pain

All participants mentioned the importance of acceptance of, and coping with their chronic pain for their ability to adapt to daily life and self-manage activities.

### Acceptance

Acceptance of chronic pain and associated limitations was a complicated process for all participants and lasted several years. Most of them did not receive professional psychological support before they became a patient of a pain clinic, which participants explained by the fact that a general practitioner or other healthcare professionals did not offer such help. On the other hand, participants did not ask for psychological support themselves. Instead, they experienced that their limitations forced them to repeatedly adjust their capabilities to lower levels.

In the early stages of their illness, they believed in and hoped for full recovery, reintegration and regaining a social life as a healthy person. Most participants mentioned a turning point

**Table 2. Categories and themes.**

| | Results |
|---|---|
| **Categories** | **Themes** |
| • Acceptation<br>• Coping | Dealing with chronic pain |
| • Consequences<br>• Bodily functioning<br>• Mental function and perception<br>• Spiritual dimension<br>• Quality of life<br>• Social and societal participation<br>• Daily functioning | The current situation regarding aspects of Positive Health |
| • Unsecure about the future<br>• Wishes and ideals | Future perspectives on health and quality of life |

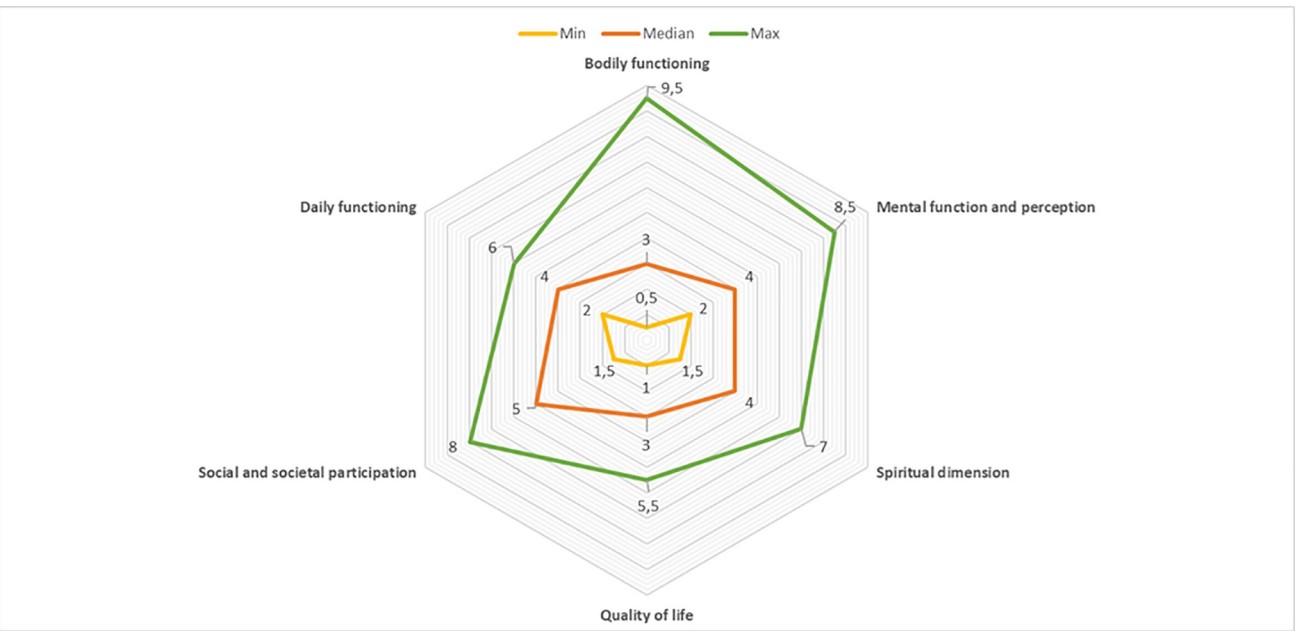

**Fig 1. Diagram of Positive Health of the study population.**

somewhere between two and six years after the start of the chronic pain and try to accept their situation and illness.

*"In the long run, now almost 2.5 years, I start to accept the pain more and more, and it actually gets easier. I should think more positively and cherish the nice things which I still can do"* (P3).

Some interviewees did not wholly accept the situation.

*"I think I have only been fighting, but at some point, you think: Okay, it is a different situation now, but this had to come from myself. Look, I cannot function in the same way I use to, but I find it very difficult to accept not being able to do more as I can do know."(P.10, the participant averts his eyes).*

*"Actually, I do not want to be where I am right now. I had to abandon many things in order to survive. I cannot overcome this just by taking it easy, but I must learn to accept what it is like now. I try to make the best of it and then you can deal with it more realistically."(P.5, the participant smiles a little).*

## Coping

All participants considered it essential to remain positive and put things in perspective in their current life and to focus on the things they still can do despite their chronic pain.

*"At the very least, I think you should try everything yourself. I have always said: giving up is not an option. Yes, that has brought me to this point. Recently my motto has become "less is more"; that is something I try to stick to."(P.5).*

Interviewees mentioned that giving up was no option, although they felt that restrictions increased over time. Instead, they tried to find solutions or alternatives for their daily short-comings. Six participants discovered walking or cycling as alternatives for the sport, while two other participants aimed to pursue their hobby or study. Even concerning holidays, when they are used to plan a family activity, four participants mentioned that they try to adapt to the capabilities of their body or be in considerable pain afterwards, while other participants did not listen to their body and suffer pain later.

*"Well, I intend to go away every day during holidays. I will not let the pain slow me down, but then I come home exhausted at the end of the morning. So be it. I will continue anyway. I temporarily put aside the pain"(P.9, the participant straightens her back).*

In general, participants did not want to place their pain problem and disabilities at the centre of their family life; instead, they wanted to remain independent and tried to be meaningful in social life at a level they could handle.

## Theme 2: The current situation regarding aspects of Positive Health

All participants stated that their low back and leg pain never completely disappeared after spinal surgery; instead, it increased over time and had changed their life.

### Physical function

They mentioned the growing immobility and impairments in walking, sitting, standing or bending and needed increasing moments of rest. Additionally, most participants reported low energy levels with continuous fatigue.

*"My current situation is terrible comparing to the activities I used to do, and the energy level I used to have. I am just not able to do those basic things anymore (work, housekeeping). Now I always say: I live in half days. I mean, at night I can only sleep half a night, and during the day I can only do half of the things I want to do. So currently, I live in half days"(P.3, the participant has tears in her eyes).*

*"I use more every day than I can refuel at night."(P.5).*

*"I am glad I have a job but sometimes I feel stretched to the limit, and feel reluctant to go to work because I ask myself: How do I get through this day with so much pain"(P.12).*

Two participants explicitly mentioned that decreasing mobility leads to the use of an electric bicycle, and about half of them frequently used the car for short rides.

Because of the immobility, participants gained weight, which in two cases had led to serve obesity and accompanying health deterioration like diabetes mellitus. Two interviewees had been extraordinarily overweight and needed a gastric bypass to lose weight. Two other interviewees were able to do sports activities with professional guidance but experienced considerable pain afterwards.

Participants also mentioned the negative impact of chronic pain on leisure activities such as family outings or pursuing their hobby.

### Sleep

Participants experienced low quality of sleep. They often awoke and experienced a restless sleep, i.e. short sleep periods and a lot of twisting and turning during the night, that caused

them the feeling of insufficient rest on waking up in the morning. As a result, they felt exhausted, languid, fatigue and often needed a short nap in the late afternoon. For one patient, the impact was enormous. She often slept separately to give her partner a good night's sleep, while another patient visited a general practitioner for sleep medication.

"*Well, some nights when the pain is less and then I sleep well for 4–5 hours or so, but there are also nights when I only slept 1–2 hours. I always say I have one good night and three bad nights, this is a pattern I developed.*" (P.5).

## Food

All participants mentioned they became more conscious of food because of their health and unwanted weight gain. However, we found that one of the main challenges of low back pain and leg pain appeared to being in a standing position e.g. during cooking. Consequently, most participants mentioned cooking and preparing food as a challenging activity or daily recurring struggle. To minimize the time in the kitchen they cooked simple dishes. When they are in much pain they feel not like cooking, and skip meals. Cooking was most difficult for the few single participants, not being able to get help from a partner or children. However participants with partners and children mentioned to feel guilty when the cooking was done by the family.

"*Well, if I'm in a lot of pain, I am not always in the mood for eating. Sometimes, I skip a meal or take something simple to eat. If I'm alone, I do not always cook. But on the better days or when my son comes home for dinner I try to cook, which takes a lot of energy. If I have a lot of pain then I eat less,*" (P.8).

## Medication

All interviewees tried various strong painkillers with ranging doses, and almost all still used them. Some patients consciously took pain medication when they had to engage in different activities. Nevertheless, all participants mentioned that despite the pain medication, they were not pain-free, but felt limited and had to monitor their boundaries. Additionally, participants mentioned an increasing dosage to reach the same analgesic effect. They also mentioned all kind of side effects, such as decreased appetite, nausea, vomiting, drowsiness, fatigue and loss of concentration (Table 1).

"*But when I am at work, I do not use pain medication because I am not going to sit at the machine with those heavy painkillers; actually, I think it is a bit dangerous.*"(P.10).

## Self-esteem

All participants reported the loss of their self-esteem over time. Mainly expressing itself as intangible and emotional changes such as, the relationship between spouses, feelings of shame towards the children and partner, lack of a paid job, a gap in the curriculum vitae, dissatisfaction with their body, but also disappointment with the results of surgery or pain treatments once they were treated at the pain clinic. One patient felt trapped in an old and worn body.

"*I should be in the prime of my life, but unfortunately, this is not what I feel at the moment. I had never expected this from the hernia. I thought it would be one operation, and then everything would be* fine."(P.13, *the participant shakes her head).

## Autonomy

Interviewees wished to maintain their independence as long as possible and indicated not quickly asking for help because this felt as shame, loss of control or loss of self-esteem.

> "And at home, if there's something to be done, no problem they will do it, you know. But, I am someone who likes to take things into my own hands, and keep it that way, even if it is difficult."(P.10).

In daily practice, participants experienced most help from their relatives at home. However, when they got help from the children, they experienced this as confronting. It caused a feeling of guilt or being side lined.

> "I am glad she is doing it, and I am glad she is a very well-raised sweet girl (6 years old) who wants to help me, but she should not have to, she should not be helping me."(P11, the partici-pant averts her eyes).

Despite sacrifices, patients emphasized the importance of being independent and make their own decisions in daily routine, although this resulted in pain and needing time to rest afterwards.

## Meaningfulness

The meaning of life for all interviewees was associated with doing something useful in family life or society; however, they struggled with the physical and psychological shortcomings.
Despite this, three interviewees mentioned the care for needy or sick parents as a useful goal. Two other participants worked as a volunteer. It provided a feeling of satisfaction to do something for other people. Some participants experienced support and strength from their faith. One interviewee, who was not a religious person, told:

> "But sometimes, I do ask God to help me get rid of my pain. And then I talk to him like that, yes, that gives me peace of mind."(P.14).

Additionally, one patient mentioned satisfaction when he could get positive things out of his daily functioning; however, this was a daily struggle and depressed him.

> "But what is the meaning of my life if I am continuously overwhelmed by pain? Then it is hard to get anything positive out of it; I find that very difficult and depressing."(P.10).

## Quality of life

All interviewees felt that they were out of balance and stated that their quality of life would improve if they were in less pain, without hindrance in activities and sleep, felt healthy, ener-getic, with a paid job, and socially active. One participant mentioned that her quality of life changed due to her pain but that the bar does not always have to be high to enjoy life.

> "Recently during holiday, I have been able to enjoy very small things. Yes, that is something I have become more aware of since the pain. I enjoy the little things more intensely because before (having chronic pain) everything was very ordinary. That is the insight I have gai-ned."(P.14, the participant smiles a little).

Eight participants also considered a holiday is contributing to the quality of life. However, they were often not able to enjoy relaxing activities. A holiday with the family at an amusement park was a valuable moment of enjoyment, but afterwards, the participant experienced severe pain.

"*I do it for my daughters, but the next day is hell because I have excruciating pain.*"(P. 11, *the participant becomes emotional*).

## Intimacy and sexuality

Participants mentioned intimacy and sexuality essential for their quality of life, but because of chronic pain, the sexual relationship with their partner had changed, and they felt less inclined to be intimate. They thought that this was not just due to the pain but also be caused by medication use, a low energy level, and continuously being fatigued.

Sexually active patients had less pleasure in Intimacy and sexuality because they experienced pain when moving. The subject was negotiable in most relationships. Although they experienced understanding and support, and there was still some intimacy and touch, participants felt a sense of shame and inadequacy.

"*I am not completely impotent, but sometimes it is just not possible because of the pain. I cannot move my body properly.*"(P.12).

"Yes, because of the pain, it does not work that well. That does something to you; it makes you feel tiny."(P.4, the participant closes his eyes).

## Social media

Social media and the internet played an essential role in the lives of participants with chronic pain. The internet was used for the world (or local) news, shopping, studying and expanding knowledge, while social media was used as a means of contact to meet other individuals with chronic pain. They used it to exchange experiences, support each other, and draw strength from this. One participant reported that this gave him a greater sense of belonging. However, on the other hand, this was also experienced as confronting.

"*If you have any questions, you talk to each other. Sometimes you think that you are the only one who is so young, but fortunately, that is not the case. Yes, you feel old when you cannot do much, but then you see on social media that there are even younger people. At least you do not feel alone.*" (P.4).

## Work

All seven participants who had a payed job considered "paid work" and "staying at work" vital as they were meaningfully engaged and actively involved in society. However, they reported that they continuously experienced pressure to perform, and therefore felt fear of making mistakes or feel difficulties in fulfilling the working day. Five interviewees, eventually had to stop working because of an increase in pain complaints, or were not given a contract extension due to frequent and/or long-term illness. One participant continued his work on a lower level, and as a result, his salary was adjusted downwards.

*"It is hard to participate in a performance-oriented company if you are limited by pain symptoms that include concentration problems."(P.5,).*

*"After 40 years of working, you have to quit. You cannot just accept it. That is not yet possible. I am too young to stop working."(P.1, the participant raises his voice).*

Several participants were disappointed with agencies help to return to work. They experienced little empathy, work that did not match with what a participant could do, or that replacement of work was short-lived.

Those participants who were entirely incapacitated, because of the chronic pain, and could barely make ends meet on their monthly salary suffered financial distress. There was neither enough commercial space for extras such as sports for the children or a family holiday, nor could they save. Interviewees indicated that they could not help ending up in this situation, which gave them a feeling of injustice. Financial problems were discussed in the domestic environment, so children understood why certain things were not possible. However, parents felt guilty about this.

*"Well, I would like to do more sports with them or go on holiday and show them more of the world, but unfortunately that is not the case now."(P.5, the participant averts his eyes).*

### Daily care and daily schedule

Two interviewees mentioned fear when showering because the bending and lifting of the legs being painful, and they were afraid to fall. They wanted their partner to stay close to this action. Patients also mentioned difficulties dressing or putting on socks and taught themselves peculiar attitudes to be able to do this independently.

All participants managed their time very carefully and weighed their activities based on the amount of energy and their rest moments. Therefore, they experienced that everything they do, take twice as long. Additionally, a high pain score is a warning that they are at their border or have already crossed it. One participant expressed it like this:

*"Chronic pain turns my life upside down, I cannot work, I cannot do many normal things, and everything is affected and limited by it, everything is now the same and boring." (P.3, the participant shakes his head).*

## Theme 3: Future perspectives on health and quality of life

Interviewees mentioned they felt insecure and anxious about their future and found it challenging to describe their plans for the future. They explained their uncertainty because they realised that the pain never goes away, limitations were increasing, and that they had to anticipate the unpredictability of pain in daily life.

*"I notice that the pain is getting worse. I find it harder to walk, and that frightens me. I wonder: where is this going?"(P.9, the participant has tears in her eyes).*

*"For me a big question mark: where is it going to end, and when and to what extent can I still do things? I do not know anything about my future, and I do not like that."(P.13).*

*"Because your body no longer does what you want to. In my mind, I want to do all kind of things, but I cannot do much. As if it does not fit together." (P.16,).*

All participants wished less pain, fewer limitations and more energy to resume their social lives and be an active part of society. They wanted to be less anxious, more in physical and emotional control, wish stability about the future and to act more spontaneous in daily activities. Two participants wished for empathy from their current work environment while two other participants wished more help in the gap in returning to the working society. Four interviewees explicitly mentioned to develop themselves with perspectives for an excellent job and a better financial situation.

## Discussion

Guided by the six dimensions of Positive Health, with in-depth qualitative interviews we explored the impact of chronic pain in seventeen Failed Back Surgery Syndrome patients, two-three weeks before they received an SCS system, on their health and their ability to adapt and self-manage in daily life.

Three themes emerged: 1) Dealing with chronic pain, 2) The current situation regarding aspects of Positive Health and 3) Future perspectives on health and quality of life.

Participants gave low scores on all six axes of the Web diagram of Positive Health.

We found that accepting and controlling chronic pain took years and is a crucial feature in patients'illness insights, influencing daily activities. Participants believed that their chronic pain is an impairment factor for the range movement and mobile disabilities, and some of the participants attempt to control their chronic pain by avoiding behaviour. However, it is known that this is not helpful: avoiding behaviour is a predictor for depressing and disappointing feelings, often because of unrealistic high expectations and being in control with their chronic pain [14]. Studies showed that this behaviour could be changed if patients give up struggling with unyielding pain and try to accept the current chronic situation, resulting in better physical and mental health [15–17]. An earlier study explained that, in the run-up to the acceptance of chronic pain, patients use a coping strategy to reduce the negative pain-related thoughts, overcome anxiety and depression sickness and simultaneously build trust and social support [18, 19]. Epstein et al. described in more detail the complicity of acceptance of chronic pain and argued that patients should accept that cure is unlikely and that they should stop looking for alternatives or avoid engagement, but on the contrary, should participate in life despite the pain [20]. This is in line with the interviewees in our study, who eventually gave up fighting and try to be socially active, meaning that they try to accept their pain by focusing on the things they still can do, and be actively involved in family and social life.

In our study, participants mentioned that poor sleep quality has an enormous impact on daily life; however, they did not consulted experts for their sleep disturbance. Indeed, a previous study showed that 50–80% of patients with chronic pain suffer from sleep problems [21, 22]. Tang et al. concluded that managing the predictors for sleep disturbance increased not only the quality of sleep but also had a beneficial effect on how patients experienced their chronic pain since pain and sleep are bidirectional. Additional, other studies argued the lack of sleep quality questions from healthcare professionals in the evaluation of chronic pain [14].

Another striking point that occurs from the interviews was that participants mentioned not to be in control with their mental function and perception for which only one participant had psychological help. The absence of goal identification in daily activities might have contributed to disturbing mental well-being [15, 23]. In these papers, it is argued that personal goals on physical and psychological levels are associated with an increased mental well-being, satisfaction and quality of care, including: problem disengagement, goal orientation, personal strength, and how the goal may be accomplished [13, 19, 23].

However, to develop personal goals, health care professionals should collaborate with a patient for adherence, motivation, and to correctly understand their needs [13, 23].

Huber et al, introduced the Web diagram of Positive Health for the broad perception of health with six dimensions preferred by the patients [8]. Patient-centred care and goal identification are the basic principles of the Positive Health diagram, as we used in our study. This model not only gave insights in patients 'personal needs, but it could also help healthcare professionals and patients with shared decision making for individual treatment in the evaluation of chronic pain resulting in a better quality of life [8].

Quality of life was also scored low in our study. Participants mentioned that a diminished Quality of life was caused by an increasing dependence in self-management, bad sleep quality and not being involved in social life. This is in line with an earlier survey from Turk et al. [24] where responders from a focus group mentioned nineteen items, which are essential in chronic pain, such as enjoyment of life, emotional well-being and sleep-related problems. Additionally to these findings, in our study participants also mentioned a religious source and a paid job as conditions for their quality of life.

Four participants mentioned that they got strength from a religious source to handle their chronic pain problems. However, from literature, we learned that the majority of patients with chronic pain have a moderate or indifferent interest in spirituality or religiosity but a firmer belief in medical control [25]. It fits most physicians who generally are more focused on pain medicine and not convinced of the benefit of spirituality or religious facts in patients' health. However, Bussing et al. stated that physicians should not be blind to the possibility of spiritual and religious beliefs and should be open for sources that encourage compliance and accountability that improve coping with chronic pain and eventually have an additional positive influence on the medical outcome [25, 26]. However, spirituality is still underexposed in regular care for patients with chronic pain.

Almost all participants mentioned that having a paid job was meaningful and vital for their quality of life. However, with increasing limitations because of the deteriorated chronic pain illness, participants had trouble to fulfilling their duties. Different papers described the importance of motivation to overcome physical limitations or personal barriers such as anxiety or uncomfortable feelings to explain work absence, or combined a job with their chronic pain disease [27–29]. Therefore, they argued that coping is the best strategy to improve independent daily functionality or work opportunities.

## Strength and limitations

We interviewed a homogeneous group of seventeen Failed Back Surgery Syndrome patients with low back pain and leg pain in the period before SCS therapy. By these open, in-depth interviews, guided by the Positive Health framework, we managed to get novel insights in what FBSS patients with chronic pain find essential in their health and ability to adapt to daily life and self-management.

The standardized pain and quality of measurements instruments, we use in daily practice in a pain clinic do not cover the complexity of a chronic pain nor explore personal feelings and needs in SCS therapy [30]. Therefore, this qualitative study is a manner of understanding the personal needs of CLBLP patients due to FBSS, regarding the six axes of the Web diagram of Positive Health. Translating these needs into personal goals can be helpful at the start of the SCS therapy as well as in the evaluation of the long-term follow-up.

Another strength is the fact that one person performed all interviews, being an experienced master of arts in advanced nursing practice in the management of patients with chronic pain and in the field of neuromodulation. She was the most adequate person to interview this

complicated group of FBSS patients and explored their personal needs and limitations in daily life. The risk of inclusion bias or any other form of bias was minimized, as: 1) patients were recruited by pain specialists and not by the interviewer, 2) the Interviewer did not know the patients before the interview, 3) the interviewer experiences' with chronic pain were used to gain more depth in the interviews, 4) all interviews were coded separately by two researchers: the second one being no clinical expert in pain management, 5) the members of the research team, who combined codes into categories and themes, were no employees of the hospital, where the patients were included, and 6) this study covered the period before patients received a neuromodulation system. Neuromodulation was not the main topic of investigation.

Final, we showed how chronic pain influenced patients'ability to adapt to daily life and self-management in this performance-oriented society. Therefore, these patients should be benefited by multidisciplinary treatment based on pain related personal factors as confirmed by the International Association for the Study of Pain (IASP) and the International Classification of Diseases (ICD-11).

The limitation in this study might be caused by the long, painful history of some of the study patients (varies from four-22 years), as it is known that patients with a long, painful history could have a bad coping strategy, a poor self-reflecting attitude and fear-avoidance behaviour [14].

## Conclusions

This study explored FBSS patients' ability to adapt to the complexity of chronic pain in daily life and showed that all participants, scored low on the pain level, and the six dimensions of the Web diagram of Positive Health. Therefore, we recommend a patient-centred approach, guided by the Positive Health model focussing on appropriate individual goals and personal needs, which could be helpful in shared decision making between a patient and a healthcare professional in the evaluation of the complexity of chronic pain.

## Supporting information

**S1 File. Coding tree.**
(XLSX)

**S2 File. Topic guide.**
(DOCX)

**S3 File. Audit trail.**
(XLSX)

**S4 File. COnsolidated criteria for REporting Qualitative research (COREQ).**
(DOCX)

## Acknowledgments

The authors would like to thank all participants for their time and contribution to this study.

## Author Contributions

**Conceptualization:** Tanja E. Hamm-Faber, Yvonne Engels, Kris C. P. Vissers, Dylan J. H. A. Henssen.

**Data curation:** Tanja E. Hamm-Faber.

**Formal analysis:** Tanja E. Hamm-Faber, Yvonne Engels, Kris C. P. Vissers, Dylan J. H. A. Henssen.

**Investigation:** Tanja E. Hamm-Faber.

**Methodology:** Tanja E. Hamm-Faber, Yvonne Engels, Kris C. P. Vissers, Dylan J. H. A. Henssen.

**Project administration:** Tanja E. Hamm-Faber.

**Resources:** Tanja E. Hamm-Faber.

**Supervision:** Yvonne Engels, Kris C. P. Vissers, Dylan J. H. A. Henssen.

**Validation:** Tanja E. Hamm-Faber, Yvonne Engels, Kris C. P. Vissers, Dylan J. H. A. Henssen.

**Visualization:** Tanja E. Hamm-Faber.

**Writing – original draft:** Tanja E. Hamm-Faber.

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
