## [Decision Letter · Decision Letter 0]

13 Oct 2020

PONE-D-20-27211

Views of patients suffering from Failed Back Surgery Syndrome 

on their health and their ability to adapt to daily life and self-management: A Qualitative Exploration

PLOS ONE

Dear Dr. Hamm-Faber,

Thank you for submitting your manuscript to PLOS ONE. After careful consideration, we feel that it has merit but does not fully meet PLOS ONE’s publication criteria as it currently stands. Therefore, we invite you to submit a revised version of the manuscript that addresses the points raised during the review process.

Please see comments below. 

We look forward to receiving your revised manuscript.

Kind regards,

Andrew Soundy

Academic Editor

PLOS ONE

Journal Requirements:

Additional Editor Comments (if provided):

Please answer the comments from the reviewer and my comments below. Failure to answer the comments could result in rejection at the next stage.

Methods

Identify that you used COREQ et al., (2007) in the text. However, when I checked it your methodological orientation and theory is incorrect – please identify a paradigmatic view and methodology. Also justify sample size. Identify what you mean as a comfortable environment – can you identify exact locations. Description of the coding tree can be given in a supplementary file please. Derivation of themes. - Did you use of a framework not focus the responses? Clarity of minor themes is applicable you identify lots of codes and should identify any negative cases.

Text

Wording line 111 “They need to master” = this is part of eligibility criteria and mentioned as exclusion? Integrate into one place

Line 120 What is a comfortable environment? Private room? Café?

Line 124 Design: identify paradigmatic view and methodological approach - may be move up front or list as per framework

Have sub-title for interview schedule

Give the interview guide in a supplementary file. Please identify if a pilot interview was done or a cognitive interview? Identify when and how many additional topics were added? How did you achieve saturation of these topics if they were found later in the process? Explain and justify with reference? Please use a supplementary file to show all this detail.

Line 140 – so you asked them why the scored 10? Or 8?

Line 144 – if necessary field notes were made? When was it necessary? Did you use any field notes in the results?

Please supply a reference for your code system used so it can be verified.

Please give an audit trail in a supplementary file

Please have a section on quality to show how quality was identified

Results

You have mentioned about comfortable environment before.

Acceptance is regarded as a part of psychological adaptation and or identified as a coping strategies, you also mention hope here – is that a part of acceptance? – both have a big literature base. The quotes says the person finds it difficult to accept or has to learn to accept. Does that mean they really accept? For me there is more to this area.

Coping - is a large area so there is lots of ways to describe this theme and codes identified. Within this section you mention numbers 5 said this, 4 said that - make sure you are consistent across results

Food – quite a small section. “They cooked simple dishes” – so all cooked? Did any have partners who cooked? How often were they alone?

Reviewers' comments:

Reviewer's Responses to Questions

**Comments to the Author**

1. Is the manuscript technically sound, and do the data support the conclusions?

Reviewer #1: Partly

2. Has the statistical analysis been performed appropriately and rigorously? 

Reviewer #1: N/A

3. Have the authors made all data underlying the findings in their manuscript fully available?

Reviewer #1: Yes

4. Is the manuscript presented in an intelligible fashion and written in standard English?

Reviewer #1: Yes

5. Review Comments to the Author

Reviewer #1: Thank you for your submission, this is a very interesting subject. I have the following comments to make

1. Methods - change to 'guided by the positive healthcare model'

2. Conclusion line 43 what do you mean by 'showed a low rate'?

3. Introduction page 3 line 72 this sentence needs a reference

4. page 4 line 93 this needs to be changed' to guided by/informed by' the elements of the Positive Health Belie model, otherwise you would be using a structured interview not semi-structured process. Please make this explicit throughout your manuscript.

5. Materials and methods - was participation voluntary and confidential? Please state

6. Page 5 lines 111-112 please change 'master the Dutch language' to a more scientific term e.g. 'needed to be fluent in the Dutch language' and 'limitations' to 'experiences' of chronic pain

7. line 113 why did participants have to meet the criteria of the Dutch Neuromodulation Society?

8. Page 5 line 121 - was data saturation or theoretical saturation reached? How did you know this?

9. Page 5 Line 127 Please give a figure of your topic guide so the reader can see this and how it was informed by the Positive Health Model

10. page 6 line 140 Why have you only mentioned this spider web now? How is this different to the topic guide?

11. Was there a theoretical or philosphical perspective that underpinned this study? If it did, then you should mention this here.

12. Page 6 line 162 - why would you repeat interviews? This is not normally done in qualitative research. Or do you mean contact participants to clarify any points? Please amend

13. Page 7 LIne 172 you state here that the participants rated the six areas of the Positive Health Model. Why is this only being brought in now. This should be introduced earlier in the methods section. How did you ensure that this would not bias the participants' responses

Page 9 Line 227 There is only one quote here, and it is very short. Other themes have longer quotes and more than one. Please give another quote, so this theme has same detail as the others

Page 10 - please supply a quote for the 'food' theme

Page 13 - Line 38 please change 'payed' to 'paid'

Page 15 line 405 please add 'informed, or guided by' the Positive Health Model

Page 16 line 424 Please clarify what you mean by 'giving up and trying to be part of society'

Page 19 line 499 If your interviewer was an expert in patient management and neuromodulation, how did he/she maintain reflexivity with the participants and data to ensure that any personal biases and assumptions were not thrust on the data?

6. PLOS authors have the option to publish the peer review history of their article (what does this mean?). If published, this will include your full peer review and any attached files.

Reviewer #1: No

---

## [Author Response · Author response to Decision Letter 0]

11 Nov 2020

Editor Comments to the author

Dear editor,

Thank you for your time to review our manuscript.

We appreciate your feedback, which helped us to improve it. Below, we react on the comments. In the revised manuscript, we highlighted the adaptations, as requested.

Methods

Identify that you used COREQ et al., (2007) in the text. However, when I checked it your methodological orientation and theory is incorrect – please identify a paradigmatic view and methodology. 

We used your comment to clarify the methodology and added the text in the methodology section under the chapter analysis. Page 8, line 181-194. It now reads:

“For the qualitative content analysis, we used the phases of thematic analysis with an inductive approach using the six dimensions of Positive Health as themes for the semi-structured in-depth interviews. (ref: Braun Virginia et al. Using thematic analysis in psychology, Qualitative research in psychology, January 2006, 3(2):77-101). 

Starting after the first interview, transcriptions were coded line-by-line, through which a code list was created. The code list derived from the previous interview was used as a starting point for coding the next one. Coding was continued after each interview. Coding and analysis were carried out independently by two researchers (T.H. and D.H.), who met periodically to compare individual coding and to discuss codes and themes until consensus was reached. By employing this methodology, the researchers independently signalled data saturation when no new topics, themes or codes arose when analysing new interviews. After signalling data saturation, two additional interviews were held to ensure no new topics or codes arose. After all interviews were conducted, transcribed and coded, the research team (T.H, Y.E, K.V. and D.H.) combined codes into categories and themes.”

Also justify sample size. 

In explorative, qualitative research, sample size continues until saturation is reached. This means that in the line by line coding process, no new codes emerge. To confirm data saturation, some additional data collection takes place. In our study, we performed two extra interviews to confirm data saturation. Page 5, line 122-124.

Identify what you mean as a comfortable environment – can you identify exact locations. 

We explained the comfortable environment and added the text. Page 5, line 135-138.

“The hospital location was an outpatient clinic room with a window view on a park. The green and white coloured room is decorated with modern art, and had two comfortable chairs for the interviewer and interviewee. On the table between them was coffee/tea and cake and the temperature was adjusted to patients `wishes”.

Description of the coding tree can be given in a supplementary file please. Derivation of themes. - Did you use of a framework not focus the responses? Clarity of minor themes is applicable you identify lots of codes and should identify any negative cases.

We added the coding tree, with categories, themes and codes as supplementary file 1. 

We reported the categories and themes in our results and displayed striking results in the discussion. 

Text

Wording line 111 “They need to master” = this is part of eligibility criteria and mentioned as exclusion? Integrate into one place.

We changed the text and integrate it in one place. Page 5, line 129-133.

“Inclusion criteria were: 1) fluency in the Dutch language 2) age ≥ 18 years, 3) diagnosed with FBSS with no option for further surgical interventions, 4) severe pain ≥ 5 on a Visual Analogue Scale for more than six months, and 5) no psychopathology (no mental illness or substance abuse) as determined by a psychological assessment.

Exclusion criteria for this study were clinical contraindications for SCS.

Line 120 What is a comfortable environment? Private room? Café?

We explained the comfortable environment and added the text under the participant chapter, in the methods and instruments section. Page 5, line 135-138. It now reads: 

“The location was an outpatient clinic room with a window view on a park. The green and white coloured room is decorated with modern art, and had two comfortable chairs for the interviewer and interviewee. On the table between them was coffee/tea and cake and the temperature was adjusted to patients `wishes”.

Line 124 Design: identify paradigmatic view and methodological approach - may be move up front or list as per framework

We explained our methodology in your first comment and added the text to the manuscript in the methods and instruments section under the chapter analysis. Page 8, line 181-194.

Have sub-title for interview schedule

We added the subtitle: “planning and interview” in the methods and instruments section. P 7, line 166.

Give the interview guide in a supplementary file.

The topic list we used for the interview guide is added as supplementary file 2.

Please identify if a pilot interview was done or a cognitive interview? 

To explain your comment, we added the text below to the Ethics statements in the methods and instruments section. Page 4, line 107-111

“The interviewer (T.H.) was trained in qualitative research and communication techniques. Furthermore, she is experienced in the management of patients suffering from chronic pain and neuromodulation therapy. We refined the interview skills of the interviewer by discussing audio recorded interviews with an experience interviewer (J.W.), and receiving feedback by two experts in scientific qualitative research (Y.E. D.H.)”. 

Identify when and how many additional topics were added?

How did you achieve saturation of these topics if they were found later in the process? Explain and justify with reference? Please use a supplementary file to show all this detail.

The text is refined on Page 7, line 160-165

We started with a topic list guided by the six dimension of the Web diagram of Positive Health. 

Per main topic we had several sub topics.

During the interview we refined these sub topics. 

The topic list is added as supplementary file 2.

Ref: Analysis in Qualitative Research, by Hennie Boeije, 2010, page 67-70 

Line 140 – so you asked them why the scored 10? Or 8?

We explained the Web diagram of Positive Health in the manuscript in the design chapter of the methods and instruments section. Page 7, line 153-159. It now reads:

“If patients were interested in taking part in the study, the Web diagram of Positive Health was explained, as a model that helps to visualise health aspects and starts a patient`s thinking process on the six dimensions of Positive Health. The patient is asked to give a score from 0 (no quality) to 10 (best quality) for each of the six dimensions. Patients had received a copy to take home during a previous outpatient clinic visit, and they completed this template on the day of the interview. This tool visualises and quantifies the six dimension of the Web diagram and initiates communication with the patient”

.

Line 144 – if necessary field notes were made? When was it necessary? Did you use any field notes in the results?

Field notes were made for elements, which could not be recorded, such as `participant became emotional or eye movements of the participants etc. If applicable, we added such a note to the quotes throughout our manuscript e.g 

Line 264: Participant averts his eyes

Line 267: Participant smiles a little

Line 285: Participant straightens her back 

Line 301: Participant has tears in her eyes: 

Other added notes are in line: 362, 375, 402, 407, 422, 447, 459, 472, 480, 

Please supply a reference for your code system used so it can be verified.

The coding was performed using ATLAS.ti. Scientific Software Development GmbH, Berlin. Document version: 673.20190310 (http://atlasti.com). Page 8, line 195-196

Please give an audit trail in a supplementary file

An audit trail is added as supplementary file 3.

Please have a section on quality to show how quality was identified.

We added a quality section on Page 8, line 199-207. It reads: 

“Quality 

The study was conducted in accordance with the principles of the Declaration of Helsinski and approved by the Medical Ethical Board. All participants voluntary signed the informed consent and receive a code, not based on patients characters, to guaranteed their privacy. 

In this qualitative study, we used audio-taped data. The transcript were analysed in Atlas.ti, under licensed of the Radboudumc, All data were anonymously stored at a Investigators Site File at the Radboudumc and only accessible by qualified control persons and research management team. Data will be saved for15 years”.

Results

You have mentioned about comfortable environment before.:

We explained the comfortable environment in your earlier remarks and added this on Page 5, line 135-138. 

Acceptance is regarded as a part of psychological adaptation and or identified as a coping strategies, you also mention hope here – is that a part of acceptance? – both have a big literature base. The quotes says the person finds it difficult to accept or has to learn to accept. Does that mean they really accept? For me there is more to this area.

The goal of our study was to explore patients ability to adapt to daily life and self-management. From our study we learned that acceptance, hope and coping are essential and meaningful issues for patients who constantly live with chronic pain and try to survive in this performance-oriented society. Therefore our interviews were guided by the Web diagram of Positive Health, focusing on the six dimensions the patients prefer. 

In the discussion we displayed the relevance of acceptance of chronic pain, resulting in a lower pain level and / or fewer mental health problems such as depression or anxiety. 

However, acceptance of chronic pain is a complicated and convoluted process and not the main issue of this study. 

Coping - is a large area so there is lots of ways to describe this theme and codes identified. Within this section you mention numbers 5 said this, 4 said that - make sure you are consistent across results

Coping is a large area. Therefore, we displayed this subject in our discussion, and stated that coping is a strategy to reduce the negative pain-related thoughts, overcome anxiety and depression sickness and simultaneously build trust and social support. 

This was recognisable from our participants, who tried to focus on the things they still can do and be part of society.

The Web diagram of Positive Health model, first introduced by Machteld et al., starts patients’ thinking process on the six dimensions of Positive Health, which could help patients to visualised their health problems and focused on personal goals and needs, and the things they still can do. 

Food – quite a small section. “They cooked simple dishes” – so all cooked? Did any have partners who cooked? How often were they alone?

We changed the text in the food section. Page 14, line 328- 340. It now reads: 

“All participants mentioned they became more conscious of food because of their health and unwanted weight gain. However, we found that one of the main challenges of low back pain and leg pain appeared to being in a standing position e.g. during cooking. Consequently, most participants mentioned cooking and preparing food as a challenging activity or daily recurring struggle. To minimize the time in the kitchen they cooked simple dishes. When they are in much pain they feel not like cooking, and skip meals. Cooking was most difficult for the few single participants, not being able to get help from a partner or children. However participants with partners and children mentioned to feel guilty when the cooking was done by the family”. 

Reviewer #1: Thank you for your submission, this is a very interesting subject. I have the following comments to make

Dear reviewer,

Thank you for your time to review this manuscript.

We appreciate your feedback, which helped us to improve it. Below, we react on the comments. In the revised manuscript, we highlighted the adaptations, as requested.

1. Methods - change to 'guided by the positive healthcare model'

We changed “using” to “guided by” throughout the text of this manuscript. Page 1, line 27.

2. Conclusion line 43 what do you mean by 'showed a low rate'? 

By 'showed a low rate' we meant that the patients experienced shortcomings in daily life within the six dimensions of the Web diagram of Positive Health. We modified the sentence in order to increase readability. Page 2, line 44-46. It now reads:

“This qualitative study explored FBSS patients ‘views on their health and the ability to adapt to daily life having complex chronic pain, and showed that patients experienced shortcomings in daily life within the six dimensions of the Web diagram of Positive Health before the SCS implant”. 

3. Introduction page 3 line 72 this sentence needs a reference

We added the reference in the introduction, Page 3, line 74.

4. Page 4 line 93 this needs to be changed' to guided by/informed by' the elements of the Positive Health Belief model, otherwise you would be using a structured interview not semi-structured process. Please make this explicit throughout your manuscript.

We changed this throughout our manuscript.

5. Materials and methods - was participation voluntary and confidential? Please state

We added this information to the revised text in the participants chapter. Page 5, line 127-128. It now reads:

“Participants voluntarily participated in this study and received a study number which guaranteed their privacy”.

6. Page 5 lines 111-112 please change 'master the Dutch language' to a more scientific term e.g. 'needed to be fluent in the Dutch language' and 'limitations' to 'experiences' of chronic pain

We adapted the wording. Page 5, line 129-133. It now reads:

“Inclusion criteria were: 1) fluency in the Dutch language 2) age ≥ 18 years, 3) diagnosed with FBSS with no option for further surgical interventions, 4) severe pain ≥ 5 on a Visual Analogue Scale for more than six months, and 5) no psychopathology (no mental illness or substance abuse) as determined by a psychological assessment”

7. line 113 why did participants have to meet the criteria of the Dutch Neuromodulation Society?

We explained the role of the Dutch Neuromodulation society in the revised Methods section Page 5, line 116-119. It now reads: 

“The Dutch Neuromodulation Society developed the guidelines for healthcare professionals in the Netherlands, who are directly involved in the neuromodulation therapy in order to monitor the quality of the treatment. Healthcare professionals should follow these rules to be able to proceed the therapy”. 

8. Page 5 line 121 - was data saturation or theoretical saturation reached? How did you know this?

We revised the paragraph “Data analysis” under the methods and instruments section to enhance clarity of the text. Also, we removed the sentence from the chapter “participants”. Page 8, line 184-194. It now reads: 

“Starting after the first interview, transcriptions were coded line-by-line, through which a code list was created. The code list derived from the previous interview was used as a starting point for coding the next one. Coding was continued after each interview.

Coding and analysis were carried out independently by two researchers (T.H. and D.H.), who met periodically to compare individual coding and to discuss codes and themes until consensus was reached. By employing this methodology, the researchers independently signalled data saturation when no new topics, themes or codes arose when analysing new interviews. After signalling data saturation, two additional interviews were held to ensure no new topics or codes arose. After all interviews were conducted, transcribed and coded, the research team (T.H, Y.E, K.V. and D.H.) combined codes into categories and themes”.

9. Page 5 Line 127 Please give a figure of your topic guide so the reader can see this and how it was informed by the Positive Health Model..

As also suggested by the editor we added the topic list as supplementary file 2 and not as a figure. 

10. Page 6 line 140 Why have you only mentioned this spider web now? How is this different to the topic guide?

It is true that the topic list and the Spider Web show considerable overlap, but they are not the same. We explained this more clearly Page 7, line 153-163. It now reads:

“If patients were interested in taking part in the study, the Web diagram was explained, as a model that helps to visualise health aspects and starts a patient`s thinking process on the six dimensions of Positive Health The patient is asked to give a score from 0 (no quality) to 10 (best quality) for each of the six dimensions. Patients had received a copy to take home during a previous outpatient clinic visit, and they completed this template on the day of the interview. This tool visualises and quantifies the six dimension of the Web diagram and initiates communication with the patient

The topic list (supplementary file 2) is used as a guide to structure the semi structured interview, and concerned exploration or matters that patients consider essential in their health and their ability to manage their lives with chronic pain. Furthermore, the topic list served as a guide for the most important topics the interviewer would like to discuss”.

11. Was there a theoretical or philosophical perspective that underpinned this study? If it did, then you should mention this here.

For the theoretical perspective that underpinned this study, we refer to the theory of Positive Health introduced by Huber et al. We added this in the manuscript in the design chapter under methods and instruments section. Page 7, line 147-153. It now reads:

“The theoretical part of the Positive Health was introduced by Huber et al. in which she stated that health is not the absence of illness but the ability to adapt to physical, emotional and social challenges, self-management and motivation for personal goals. She developed the Spider Web of Positive Health with the six dimensions of Positive Health, which are: 1) Bodily functioning, 2) Mental function and perception, 3) Spiritual dimension, 4) Quality of life, 5) Social and societal participation and 6) Daily functioning.”

12. Page 6 line 162 - why would you repeat interviews? This is not normally done in qualitative research. Or do you mean contact participants to clarify any points? Please amend

The sentence was misinterpreted; we did not interview participants twice. When any uncertainties would remain after analysing the interviews, the researchers would prefer to have the opportunity to re-contact the interviewees. 

We added the information in the design chapter of the methods and instruments section and removed it from result section Page 8, line 177-178. It now reads: 

“If any uncertainties would remain after analysing the interviews, the researchers had the opportunity to re-contact the interviewees”.

13. Page 7 Line 172 you state here that the participants rated the six areas of the Positive Health Model. Why is this only being brought in now. This should be introduced earlier in the methods section. How did you ensure that this would not bias the participants' responses

We replaced the text of the Web diagram of Positive Health to the begin of the design chapter in the methods an instruments section. Page 6, line 144-146. 

How did you ensure that this would not bias the participants' responses.

We clarify this in the text, Page 7, 166-173. where it reads: 

“All interviews took place 2-3 weeks before the planned SCS implantation date and were conducted by one researcher, who is trained in qualitative research and interviewing (T.H). 

The interviewer conducted the interviews as near as normal conversation by using open-ended questions. Participants were encouraged to express their feelings and to elaborate on aspects of their daily lives and possible daily challenges they encountered on all dimensions. When answers were unclear to the interviewer, additional questions were asked to ensure a clear understanding of the meaning of the patients ‘expressions”.

14. Page 9 Line 227 There is only one quote here, and it is very short. Other themes have longer quotes and more than one. Please give another quote, so this theme has same detail as the others

We added two more quotes on Page 13, line 298- 305. 

“My current situation is terrible comparing to the activities I used to do, and the energy level I used to have. I am just not able to do those basic things anymore (work, housekeeping). Now I always say: I live in half days. I mean, at night I can only sleep half a night, and during the day I can only do half of the things I want to do. So currently, I live in half days ”(P.3).

“I am glad I have a job but sometimes I feel stretched to the limit, and feel reluctant to go to work because I ask myself: How do I get through this day with so much pain ”(P.12).

15. Page 10 - please supply a quote for the 'food' theme

We added a quote for the ‘food’ theme in the result section, Page 14, line 337-340, where it now reads:

"Well, if I'm in a lot of pain, I am not always in the mood for eating. Sometimes, I skip a meal or take something simple to eat. If I am alone, I do not always cook. But on the better days or when my son comes home for dinner I try to cook, which takes a lot of energy. If I have a lot of pain then I eat less," (P.8).

16. Page 13 - Line 38 please change 'payed' to 'paid'

The typographical error was corrected on page 18, line 446.

17. Page 15 line 405 please add 'informed, or guided by' the Positive Health Model

We changed the wording throughout the text.

18. Page 16 line 424 Please clarify what you mean by 'giving up and trying to be part of society'

This has been explained now more clearly in the text on page 21, line 515-517 

Interviewees in our study eventually gave up fighting. They try to be socially active, meaning that they try to accept their pain by focusing on the things they still can do, and be actively involved in family and social life.

To clarify this, we added a quote in the acceptance chapter page 11, line 257-259, 

"In the long run, now almost 2.5 years, I start to accept the pain more and more, and it actually gets easier. I should think more positively and cherish the nice things which I still can do" (P3).

19. Page 19 line 499 If your interviewer was an expert in patient management and neuromodulation, how did he/she maintain reflexivity with the participants and data to ensure that any personal biases and assumptions were not thrust on the data?

We adapted the text in the strength and limitations section. Page 23, line 581-589 where it now reads: 

“The risk of inclusion bias or any other form of bias was minimized, as: 1) patients were recruited by pain specialists and not by the interviewer, 2) the Interviewer did not know the patients before the interview, 3) the interviewer experiences’ with chronic pain were used to gain more depth in the interviews, 4) all interviews were coded separately by two researchers: the second one being no clinical expert in pain management, 5) the members of the research team, who combined codes into categories and themes, were no employees of the hospital, where the patients were included, and 6) this study covered the period before patients received a neuromodulation system. Neuromodulation was not the main topic of investigation”.

---

## [Decision Letter · Decision Letter 1]

19 Nov 2020

Views of patients suffering from Failed Back Surgery Syndrome 

on their health and their ability to adapt to daily life and self-management: A Qualitative Exploration

PONE-D-20-27211R1

Dear Dr. Hamm-Faber,

We’re pleased to inform you that your manuscript has been judged scientifically suitable for publication and will be formally accepted for publication once it meets all outstanding technical requirements.

Kind regards,

Andrew Soundy

Academic Editor

PLOS ONE

Additional Editor Comments (optional):

Reviewers' comments:

Reviewer's Responses to Questions

**Comments to the Author**

1. If the authors have adequately addressed your comments raised in a previous round of review and you feel that this manuscript is now acceptable for publication, you may indicate that here to bypass the “Comments to the Author” section, enter your conflict of interest statement in the “Confidential to Editor” section, and submit your "Accept" recommendation.

Reviewer #1: All comments have been addressed

2. Is the manuscript technically sound, and do the data support the conclusions?

Reviewer #1: Yes

3. Has the statistical analysis been performed appropriately and rigorously? 

Reviewer #1: N/A

4. Have the authors made all data underlying the findings in their manuscript fully available?

Reviewer #1: Yes

5. Is the manuscript presented in an intelligible fashion and written in standard English?

Reviewer #1: Yes

6. Review Comments to the Author

Reviewer #1: Thank you for your revised manuscript and detailed comments. You have addressed all my suggestions in detail, many thanks

7. PLOS authors have the option to publish the peer review history of their article (what does this mean?). If published, this will include your full peer review and any attached files.

Reviewer #1: No

---

## [Editor Report · Acceptance letter]

23 Nov 2020

PONE-D-20-27211R1 

Views of patients suffering from Failed Back Surgery Syndrome on their health and their ability to adapt to daily life and self-management: A qualitative Exploration 

Dear Dr. Hamm-Faber:

I'm pleased to inform you that your manuscript has been deemed suitable for publication in PLOS ONE. Congratulations! Your manuscript is now with our production department. 

Kind regards, 

on behalf of

Dr. Andrew Soundy 

Academic Editor

PLOS ONE